# USER INFERENCE ATTACKS ON LARGE LANGUAGE MODELS

## ABSTRACT

Fine-tuning is a common and effective method for tailoring large language models (LLMs) to specialized tasks and applications. In this paper, we study the privacy implications of fine-tuning LLMs on user data. To this end, we consider a realistic threat model, called *user inference*, wherein an attacker infers whether or not a user's data was used for fine-tuning. We implement attacks for this threat model that require only a small set of samples from a user (possibly different from the samples used for training) and black-box access to the fine-tuned LLM. We find that LLMs are susceptible to user inference attacks across a variety of fine-tuning datasets, at times with near perfect attack success rates. Further, we investigate which properties make users vulnerable to user inference, finding that outlier users (i.e. those with data distributions sufficiently different from other users) and users who contribute large quantities of data are most susceptible to attack. Finally, we explore several heuristics for mitigating privacy attacks. We find that interventions in the training algorithm, such as batch or per-example gradient clipping and early stopping fail to prevent user inference. However, limiting the number of fine-tuning samples from a single user can reduce attack effectiveness, albeit at the cost of reducing the total amount of fine-tuning data.[1]

## 1 INTRODUCTION

Successfully applying large language models (LLMs) to real-world problems is often best achieved by fine-tuning on domain-specific data (Liu et al., 2022; Mosbach et al., 2023). This approach is seen in a variety of commercial products deployed today, e.g., GitHub Copilot (Chen et al., 2021), Gmail Smart Compose (Chen et al., 2019), GBoard (Xu et al., 2023), etc., that are based on LMs trained or fine-tuned on domain-specific data collected from users. The practice of fine-tuning on user data—particularly on sensitive data like emails, texts, or source code—comes with privacy concerns, as LMs have been shown to leak information from their training data (Carlini et al., 2021), especially as models are scaled larger (Carlini et al., 2023). In this paper, we study the privacy risks posed to users whose data are leveraged to fine-tune LLMs.

Most existing privacy attacks on LLMs can be grouped into two categories: *membership inference*, in which the attacker obtains access to a sample and must determine if it was trained on (Mireshghallah et al., 2022; Mattern et al., 2023; Niu et al., 2023); and *extraction attacks*, in which the attacker tries to reconstruct the training data by prompting the model with different prefixes (Carlini et al., 2021; Lukas et al., 2023). These threat models make no assumptions about the training data and thus cannot estimate the privacy risk to a user when that user contributes many, likely correlated, training samples. To this end, we consider the threat model of *user inference* (Miao et al., 2021; Hartmann et al., 2023), a realistic privacy attack for models trained on user data, in the context of LLMs.

As depicted in Figure 1, the attacker's goal in user inference is to determine if a particular user participated in LLM fine-tuning using only black-box access to the fine-tuned model and a small set of i.i.d. samples from the user. Importantly, these samples need not be part of the fine-tuning set. This threat model lifts the concept of membership inference from privacy of individual samples to privacy of users who contribute multiple samples, while also relaxing the stringent assumption that the attacker has access to samples from the fine-tuning dataset. By itself, user inference could

---

[1]Notable changes made during the rebuttal are highlighted in blue.

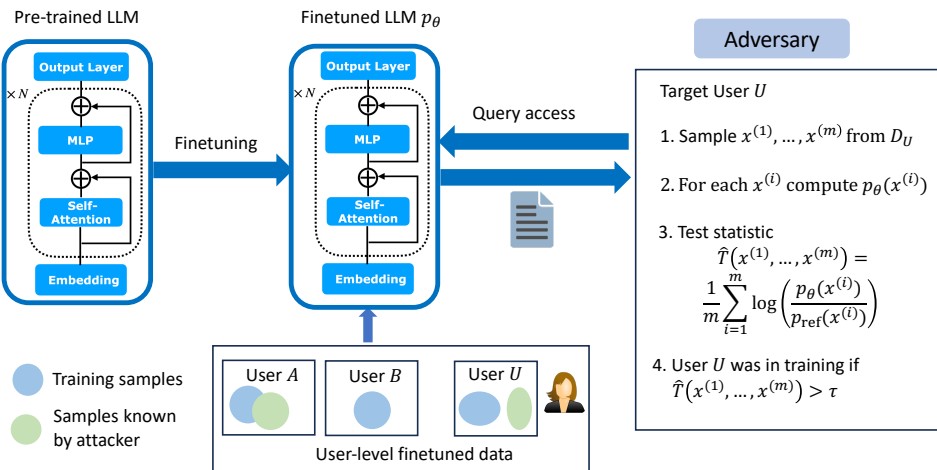

**Figure 1:** Overview of user inference threat model. An LLM model is fine-tuned on user-stratified data. The adversary can query text samples on the fine-tuned model and compute likelihoods. The adversary has knowledge of several samples from a user's distribution (different than the user training samples) and computes a likelihood score to determine if the user participated in training.

be a privacy threat if the fine-tuning task reveals sensitive information about participating users (for instance, if a model is fine-tuned only on users with a rare disease). Moreover, user inference may also enable other attacks such as sensitive information extraction, similarly to how membership inference is used as a subroutine in training data extraction attacks (Carlini et al., 2021).

In this paper, we formally define the user inference threat model and propose a practical attack that determines if a user participated in fine-tuning by computing a likelihood ratio test statistic normalized relative to a reference model (Section 3). We then empirically study the effectiveness of this attack on the GPT-Neo family of LLMs (Black et al., 2021) when fine-tuned on a diverse variety of domain-specific data, including emails, scientific writing, and news articles (Section 4.2). Our investigation gives insight into various parameters that affect how easily a user's participation can be inferred—parameters such as uniqueness of a user's data distribution, amount of fine-tuning data contributed by a user, and amount of attacker knowledge about a user.

Furthermore, we evaluate the attack on synthetically generated canary users to characterize the privacy leakage for worst-case users (Section 4.3). We show that canaries generated via minimal modifications to the real data distribution increase the attack's effectiveness by more than $40\%$ in terms of attack AUROC. Importantly, this canary study indicates that simple features shared across a user's samples, such as an email signature or short characteristic phrase, can exacerbate user inference.

Finally, we evaluate several methods for mitigating privacy attacks, such as per-example or batch gradient clipping, early stopping, and limiting the number of samples a user can contribute to the fine-tuning set (Section 4.4). We find that interventions in the training algorithm, like gradient clipping and early stopping fail to mitigate user inference, but limiting user contribution reduces the attack's effectiveness on both real and synthetic canary users. Based on these results, we highlight the importance of future work on user-level differential privacy to mitigate user inference (McMahan et al., 2018; Levy et al., 2021). Overall, our work is the first to study user inference attacks against LLMs and provides key insights to inform future deployments of LLMs fine-tuned on user data.

## 2 RELATED WORK

Over the years, a range of ML privacy attacks with different objectives have been studied (Oprea & Vassilev, 2023): *membership inference* attacks determine if a particular data sample was part of a model's training set (Shokri et al., 2017; Yeom et al., 2018; Carlini et al., 2022; Ye et al., 2022; Watson et al., 2022; Choquette-Choo et al., 2021; Jagielski et al., 2023a); *data reconstruction* aims to exactly reconstruct the training data of a model (typically for a discriminative model) (Haim et al.,

2022); and *extraction* attacks aim to extract training data from generative models like LLMs (Carlini et al., 2021; Lukas et al., 2023; Ippolito et al., 2023; Anil et al., 2023; Kudugunta et al., 2023).

**Membership inference attacks on LLMs**. Mireshghallah et al. (2022) introduce a likelihood ratio-based attack on LLMs, designed for masked language models, such as BERT. Mattern et al. (2023) compare the likelihood of a sample against the average likelihood of a set of neighboring samples, and eliminate the assumption of attacker knowledge of the training distribution used in other membership inference attacks. Debenedetti et al. (2023) study how systems built on LLMs may amplify membership inference. Carlini et al. (2021) use a perplexity-based membership inference attack to extract training data from GPT-2. Their attack prompts the LLM to generate sequences of text, and then uses membership inference to identify sequences copied from the training set. Note that membership inference requires access to exact training samples while user inference does not.

**Extraction attacks**. Following Carlini et al. (2021), memorization in LLMs received much attention (Zhang et al., 2021; Tirumala et al., 2022; Biderman et al., 2023; Anil et al., 2023). These works found that memorization scales with model size (Carlini et al., 2023) and data repetition (Kandpal et al., 2022), may eventually be forgotten (Jagielski et al., 2023b), and can exist even on models trained for specific restricted use-cases like translation (Kudugunta et al., 2023). Lukas et al. (2023) develop techniques to extract PII information from LLMs and (Inan et al., 2021) design metrics to measure how much of user's confidential data is leaked by the LLM. Once a user's participation is identified by user inference, these techniques can be used to estimate the amount of privacy leakage.

**User-level membership inference**. Much prior work on inferring whether a user's data was part of the training set makes the stronger assumption that the attacker has access to a user's exact training samples — we refer to this as *user-level membership inference* to distinguish it from *user inference* (which does not require access to the exact training samples). Song & Shmatikov (2019) give the first such an attack for generative text models. Their attack is based on training multiple shadow models and does not scale to LLMs due to its high computational cost. This threat model has also been studied for text classification via reduction to membership inference (Shejwalkar et al., 2021).

**User inference**. Finally, the user inference threat model has also been considered for speech recognition in IoT devices (Miao et al., 2021), representation learning for vision (Li et al., 2022) and face recognition (Chen et al., 2023). Hartmann et al. (2023) formally define user inference for classification and regression problems, under the name *distributional membership inference*. These attacks are either domain-specific or require shadow models and do not apply/scale to LLMs. Instead, we design an efficient user inference attacks that scale to LLMs and illustrate the user-level privacy risks posed by fine-tuning on user-generated text data. We refer to Appendix B for a detailed discussion.

## 3 USER INFERENCE ATTACKS

Consider an autoregressive language model $p_\theta$ that defines a distribution $p_\theta(x_t|\boldsymbol{x}_{<t})$ over the next token $x_t$ in continuation of a prefix $\boldsymbol{x}_{<t} = (x_1, \ldots, x_{t-1})$. We are interested in a setting where a pretrained LLM $p_{\theta_0}$ with initial parameters $\theta_0$ is fine-tuned on a dataset $D_{\mathsf{FT}}$ sampled i.i.d. from a distribution $\mathcal{D}_{\mathsf{task}}$. The most common objective is to minimize the cross entropy of predicting each next token $x_t$ given the context $\boldsymbol{x}_{<t}$ for each fine-tuning sample $\boldsymbol{x} \in D_{\mathsf{FT}}$. Thus, the fine-tuned model $p_\theta$ is trained to maximize the log-likelihood $\sum_{\boldsymbol{x} \in D_{\mathsf{FT}}} \log p_\theta(\boldsymbol{x}) = \sum_{\boldsymbol{x} \in D_{\mathsf{FT}}} \sum_{t=1}^{|\boldsymbol{x}|} \log p_\theta(x_t|\boldsymbol{x}_{<t})$ of the fine-tuning set $D_{\mathsf{FT}}$.

**Fine-tuning with user-stratified data**. Much of the data used to fine-tune LLMs has a user-level structure. For example, emails, messages, and blog posts can reflect the specific characteristics of the user who wrote them. Two text samples from the same user are more likely to be similar to each other than samples across users in terms of language use, vocabulary, context, and topics.

To capture user-stratification, we model the fine-tuning distribution $\mathcal{D}_{\mathsf{task}}$ as a mixture

$$\mathcal{D}_{\mathsf{task}} = \sum_{u=1}^{n} \alpha_u \mathcal{D}_u \qquad (1)$$

of $n$ user data distributions $\mathcal{D}_1, \ldots, \mathcal{D}_n$ with non-negative weights $\alpha_1, \ldots, \alpha_n$ that sum to one. One can sample from $\mathcal{D}_{\mathsf{task}}$ by first sampling a user $u$ with probability $\alpha_u$ and then sampling a document $\boldsymbol{x} \sim \mathcal{D}_u$ from the user's data distribution. We note that the fine-tuning process of the LLM is oblivious to user-stratification of the data.

**The user inference threat model**. The task of membership inference assumes that an attacker has full access to a text sample $x$ and must determine whether $x$ was a part of the training or fine-tuning data (Shokri et al., 2017; Yeom et al., 2018; Carlini et al., 2022). We relax this assumption on the knowledge of an attacker by considering a realistic threat model called **user inference**.

Given access to $m$ i.i.d. samples $x^{(1)}, \dots, x^{(m)} \sim \mathcal{D}_u$ from user $u$'s distribution, the task of the adversary is to determine if *any* data from user $u$ was involved in fine-tuning the model $p_\theta$. Crucially, we allow $x^{(i)} \notin D_{\mathsf{FT}}$, i.e., the attacker is not assumed to have access to the exact samples of user $u$ that were a part of the fine-tuning set. For instance, if an LLM is fine-tuned on user emails, the attacker can reasonably be assumed to have access to *some* emails from a user, but not necessarily the ones used to fine-tune the model. We believe this is a realistic threat model for LLMs, as it does not require exact knowledge of training set samples, as in membership inference attacks.

In terms of the adversarial capabilities, we assume that the attacker has *black-box access* to the LLM $p_\theta$ — they can only query the model's likelihood on a sequence of tokens and might not have knowledge of either the model architecture or parameters. Following standard practice in membership inference (Mireshghallah et al., 2022; Watson et al., 2022), we allow the attacker access to a reference model $p_{\mathsf{ref}}$ that is similar to the target model $p_\theta$ but has not been trained on user $u$'s data. This can simply be the pre-trained model $p_{\theta_0}$ or another LLM.

**Attack strategy**. The attacker's task can be formulated as a statistical hypothesis test. Letting $\mathcal{P}_u$ denote the set of models trained on user $u$'s data, the attacker's goal is to decide between:

$$H_0 \,:\, p_\theta \notin \mathcal{P}_u, \qquad H_1 \,:\, p_\theta \in \mathcal{P}_u\,. \qquad (2)$$

There is generally no prescribed recipe to test for a composite hypothesis corresponding to a set of models. The model likelihood is a natural test statistic as we might expect $p_\theta(x^{(i)})$ to be high if $H_1$ is true and low otherwise. Unfortunately, this is not always true, even for membership inference. Indeed, $p_\theta(x)$ can be large for $x \notin D_{\mathsf{FT}}$ for easy-to-predict $x$ (e.g., generic text using common words), while $p_\theta(x)$ can be small even if $x \in D_{\mathsf{FT}}$ for hard-to-predict $x$. This necessitates the need for calibrating the test using a reference model (Mireshghallah et al., 2022; Watson et al., 2022).

Our insight for designing an efficient attack strategy is to formalize the attacker's task with simpler surrogate hypotheses that are easier to test:

$$H_0' \,:\, x^{(1)}, \dots, x^{(m)} \sim p_{\mathsf{ref}}\,, \qquad H_1' \,:\, x^{(1)}, \dots, x^{(m)} \sim p_\theta\,. \qquad (3)$$

By construction, $H_0'$ is always false since $p_{\mathsf{ref}}$ is not fine-tuned on user $u$'s data. However, $H_1'$ is more likely to be true if the user $u$ participates in training *and* the samples contributed by $u$ to the fine-tuning dataset $D_{\mathsf{FT}}$ are similar to the samples $x^{(1)}, \dots, x^{(m)}$ known to the attacker even if they are not identical. In this case, the attacker rejects $H_0'$. Conversely, if user $u$ did not participate in fine-tuning and no samples from $D_{\mathsf{FT}}$ are similar to $x^{(1)}, \dots, x^{(m)}$, then the attacker finds both $H_0'$ and $H_1'$ to be equally (im)plausible, and fails to reject $H_0'$. Intuitively, to faithfully test $H_0$ vs. $H_1$ using $H_0'$ vs. $H_1'$, we require that a sample $x \sim \mathcal{D}_u$ is more similar *on average* to any other sample from the same user $x' \sim \mathcal{D}_u$ than to a sample from another user $x'' \sim \mathcal{D}_{u'}$ for any other $u' \neq u$.

The Neyman-Pearson lemma tells us that the *likelihood ratio test* is the most powerful for testing $H_0'$ vs. $H_1'$, i.e., it achieves the best true positive rate at any given false positive rate (e.g., Lehmann et al., 1986, Thm. 3.2.1). This involves constructing a test statistic using the log-likelihood ratio

$$T(x^{(1)}, \dots, x^{(m)}) := \log\left(\frac{p_\theta(x^{(1)}, \dots, x^{(m)})}{p_{\mathsf{ref}}(x^{(1)}, \dots, x^{(m)})}\right) = \sum_{i=1}^{m} \log\left(\frac{p_\theta(x^{(i)})}{p_{\mathsf{ref}}(x^{(i)})}\right)\,, \qquad (4)$$

where the last equality follows from the independence of each $x^{(i)}$, which we assume. Given a threshold $\tau$, the attacker rejects the null hypothesis and declares that $u$ has participated in fine-tuning if $T(x^{(1)}, \dots, x^{(m)}) > \tau$. In practice, the number of samples $m$ available to the attacker might vary for each user, so we normalize the statistic by $m$. Thus, our final attack statistic is the empirical mean $\hat{T}(x^{(1)}, \dots, x^{(m)}) = \frac{1}{m} T(x^{(1)}, \dots, x^{(m)})$.

**Analysis of the attack statistic**. We analyze this attack statistic in a simplified setting to gain some intuition on when we can infer the participation of user $u$. In the large sample limit as $m \to \infty$, the

mean statistic $\hat{T}$ approximates the population average

$$\bar{T}(\mathcal{D}_u) := \mathbb{E}_{\boldsymbol{x} \sim \mathcal{D}_u} \left[ \log \left( \frac{p_\theta(\boldsymbol{x})}{p_{\mathsf{ref}}(\boldsymbol{x})} \right) \right] . \tag{5}$$

We will analyze this test statistic for the choice $p_{\mathsf{ref}} = \mathcal{D}_{-u} \propto \sum_{u' \neq u} \alpha_{u'} \mathcal{D}_{u'}$, which is the fine-tuning mixture distribution excluding the data of user $u$. This is motivated by the results of Watson et al. (2022) and Sablayrolles et al. (2019), who show that using a reference model trained on the whole dataset excluding a single sample approximates the optimal membership inference classifier.

Let $\mathrm{KL}(\cdot \| \cdot)$ and $\chi^2(\cdot \| \cdot)$ denote the Kullback–Leibler and chi-squared divergences respectively. We establish the following bound, assuming $p_\theta$ and $p_{\mathsf{ref}}$ perfectly capture their target distributions.

**Proposition 1.** *Assume $p_\theta = \mathcal{D}_{\mathsf{task}}$ and $p_{\mathsf{ref}} = \mathcal{D}_{-u}$ for some user $u \in [n]$. Then, we have*

$$\log(\alpha_u) + \mathrm{KL}(\mathcal{D}_u \| \mathcal{D}_{-u}) < \bar{T}(\mathcal{D}_u) \leq \alpha_u \chi^2(\mathcal{D}_u \| \mathcal{D}_{-u}).$$

The upper and lower bounds, proved in Appendix A, provide two intuitive insights. Two types of users are susceptible to user inference:

(a) users who contribute more data to to fine-tuning (such that $\alpha_u$ is large), or
(b) users who contribute unique data (such that $\mathrm{KL}(\mathcal{D}_u \| \mathcal{D}_{-u})$ and $\chi^2(\mathcal{D}_u \| \mathcal{D}_{-u})$ are large).

Conversely, if neither condition holds, then a user's participation in fine-tuning cannot be reliably detected. Our experiments later corroborate these observations; we use them to design mitigations.

## 4 EXPERIMENTS

In this section, we empirically study the susceptibility of models to user inference attacks, the factors that affect attack performance, and potential mitigation strategies.

| Dataset | User Field | #Users | #Examples | Percentiles of Examples/User | | | | |
|---|---|---|---|---|---|---|---|---|
| | | | | $\mathbf{P_0}$ | $\mathbf{P_{25}}$ | $\mathbf{P_{50}}$ | $\mathbf{P_{75}}$ | $\mathbf{P_{100}}$ |
| ArXiv Abstracts | Submitter | 16511 | $625K$ | 20 | 24 | 30 | 41 | 3204 |
| CC News | Domain Name | 2839 | $660K$ | 30 | 50 | 87 | 192 | 24480 |
| Enron Emails | Email Address | 150 | $491K$ | 150 | 968 | 1632 | 3355 | 28229 |

**Table 1: Evaluation dataset summary statistics**: The three evaluation datasets vary in their notion of "user" (i.e. an ArXiv abstract belongs to the user who submitted it to ArXiv whereas a CC News article belongs to the web domain where the article was published). Additionally, these datasets span multiple orders of magnitude in terms of number of users and number of examples contributed per user.

### 4.1 EXPERIMENTAL SETUP

**Datasets**. We evaluate user inference attacks on three user-stratified text datasets: ArXiv Abstracts (Clement et al., 2019) for scientific paper abstracts, CC News (Hamborg et al., 2017; Charles et al., 2023) for news articles, and Enron Emails (Klimt & Yang, 2004) for real-world emails. These datasets provide a diverse test bench not only in their domain, but also in the notion of a user, the number of distinct users, and the amount of data contributed per user; see Table 1.

To make these datasets suitable for evaluating user inference attacks, we split them into a held-in set of users, that we use to fine-tune models, and a held-out set of users that we use to evaluate attacks. We set aside 10% of a user's sample as the attacker's knowledge to run user inference attacks; these samples are not used for fine-tuning. For more details on the dataset preprocessing, see Appendix C.

**Models**. We evaluate user inference attacks on the 125M and 1.3B parameter decoder-only LMs from the GPT-Neo (Black et al., 2021) model suite. These models were pre-trained on The Pile dataset (Gao et al., 2020), an 825 GB diverse text corpus, and use the same architecture and pre-training objectives as the GPT-2 and GPT-3 models. Further details on how we fine-tune these models are given in Appendix C.

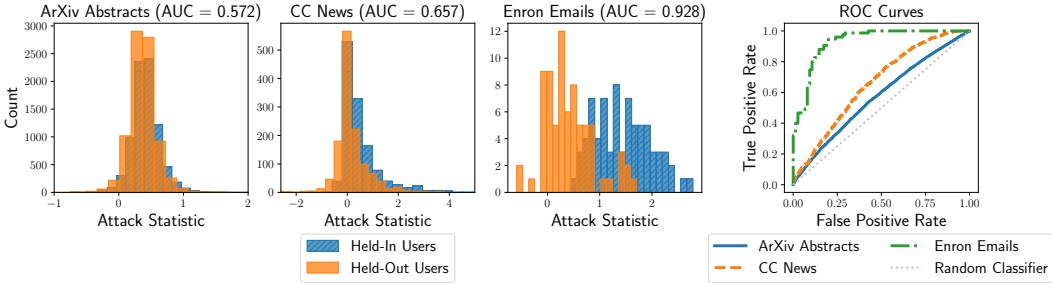

**Figure 2: Our attack can achieve significant AUROC,** e.g., on the Enron emails dataset. We show two metrics of attack performance. **(Left three)**: histograms of the test statistics for held-in and held-out users for the three attack evaluation datasets. **(Rightmost)**: Their corresponding ROC curves.

Due to the size of The Pile, we found it challenging to find user-stratified datasets that were not part of model pre-training; this is a problem with LLMs in general (Sainz et al., 2023). However, we believe that our setup still faithfully evaluates the fine-tuning setting for two main reasons. First, the overlapping fine-tuning data constitutes only a small fraction of all the data in The Pile. Second, our attacks are likely only weakened (and thus, underestimate the true risk) by this setup. This is because inclusion of the held-out users in pre-training should only reduce the model's loss on these samples, making the loss difference smaller and thus our attack harder to employ.

**Attack Setup and Evaluation**. We implement the user inference attack described in Section 3 using the pre-trained GPT-Neo models as our reference models $p_{\text{ref}}$. We evaluate the aggregate attack success using the Receiver Operating Characteristic (ROC) curve across held-in and held-out users; this is a plot of the true positive and false positive rates of the attack across all possible thresholds. We use the area under this curve (AUROC) as a single-number summary. This metric is commonly used to evaluate the performance of membership inference attacks (Carlini et al., 2022).

### 4.2 USER INFERENCE: RESULTS AND PROPERTIES

We experimentally examine how user inference is impacted by factors such as the amount of user data and attacker knowledge, the model scale, as well as the connection to overfitting.

**Attack Performance**. We begin by attacking GPT-Neo 125M trained on each of the three fine-tuning datasets and evaluating the attack performance. We see from Figure 2 that the user inference attacks on all three datasets achieve non-trivial performance, with the attack AUROC varying between 92% (Enron Emails) to 66% (CC News) and 57% (ArXiv Abstracts).

The disparity in performance between the three datasets can be explained in part by the intuition from Proposition 1, which points out two factors. First, a larger fraction of data contributed by a user makes user inference easier. The Enron dataset has fewer users, each of whom contributes a significant fraction of the fine-tuning data (cf. Table 1), while, the ArXiv dataset has a large number of users, each with few datapoints. Second, distinct user data makes user inference easier. Enron emails are more distinct due to identifying information such as names (in salutations and signatures) and addresses, while the scientific writing style of ArXiv abstracts, which is predominantly impersonal and formal, makes them less distinct.

**The Effect of the Attacker Knowledge**. We examine the effect of the attacker knowledge, i.e., the amount of user data used by the attacker to compute the test statistic, in Figure 3. First, we find that greater attacker knowledge leads to higher attack AUROC and lower variance on the attack success. For CC News, the AUROC increases from $62.0 \pm 3.3\%$ when the attacker has only one document to $68.1 \pm 0.6\%$ when the attacker has 50 documents. We also observe that the user inference attack already leads to non-trivial results with an attacker knowledge of *one document per user* for CC News (AUROC 62.0%) and Enron Emails (AUROC 73.2%). This performance for ArXiv Abstracts is, however, not much better than random (AUROC 53.6%). Overall, the results show that an attacker does not need much data to mount a strong attack, but more data only helps.

**User Inference and User-level Overfitting**. It is well-established that overfitting to the training data is sufficient for successful membership inference (Yeom et al., 2018). We find that a similar

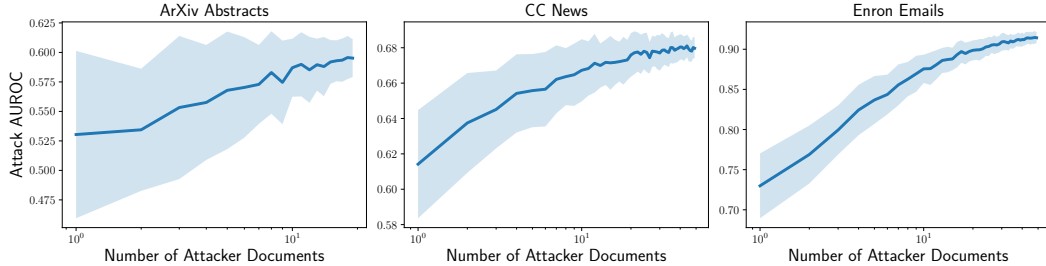

**Figure 3: Attack performance with increasing attacker knowledge**: As we increase the number of examples given to the attacker, the attack performance increases across all three datasets. At each level of attacker knowledge, we shade the AUROC standard deviation over 100 random draws of attacker examples.

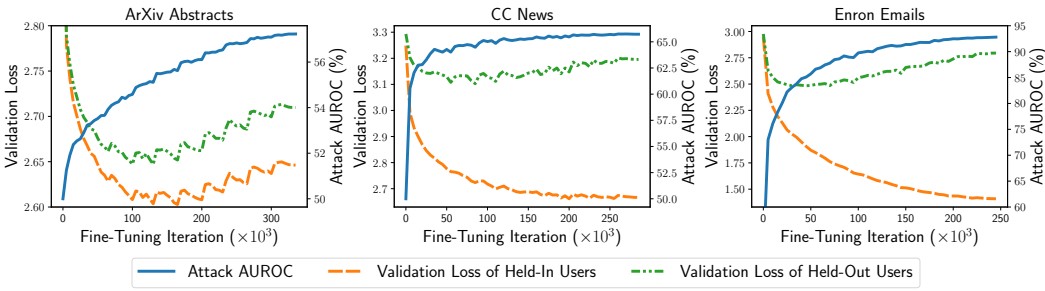

**Figure 4: Attack performance over fine-tuning**: User inference attack AUROC as well as the validation perplexity on held-in and held-out users over the course of a fine-tuning run.

phenomenon holds for user inference, which is enabled by *user-level overfitting*, i.e., the model overfits not to the training samples themselves, but rather the *distributions* of the training users.

We see from Figure 4 that the validation loss of held-in users continues to decrease for CC News and Enron Emails, while the loss of held-out users increases. These curves display a textbook example of overfitting, not to the training data (since both curves are computed using validation data), but to the distributions of the training users. We can see that the attack AUROC improves with the widening generalization gap between these two curves. Indeed, the Spearman correlation between the generalization gap and the attack AUROC is at least $99.4\%$ for *all three datasets* including ArXiv, where the trend is not as clear visually. This demonstrates the close relation between user-level overfitting and user inference.

**Attack Performance and Model Scale**. Next, we investigate the role of model scale in user inference. We fine-tune GPT-Neo 125M and 1.3B on CC News and evaluate attack performance.

We see from Figure 5, that the attack performance is nearly identical on both models with AUROCs of $65.3\%$ for the 1.3B model and $65.8\%$ for the 125M model. While the 1.3B parameter model achieves better validation loss on both held-in users (2.24 vs. 2.64) and held-out users (2.81 vs. 3.20), the generalization gap is nearly the same for both models (0.57 vs. 0.53). This shows a qualitative difference between user inference and membership inference, where in the latter threat model attack performance reliably increases with model size (Carlini et al., 2023; Tirumala et al., 2022; Kandpal et al., 2022; Mireshghallah et al., 2022).

## 4.3 USER INFERENCE IN THE WORST-CASE

The disproportionately large downside to privacy leakage necessitates looking beyond the average-case privacy risk to worst-case settings. To this end, we analyze attack performance on datasets containing synthetically generated users, known as *canaries*. There is usually a trade-off between making the canary users realistic and worsening their privacy risk. We intentionally err on the side of making them realistic to illustrate the potential risks of user inference.

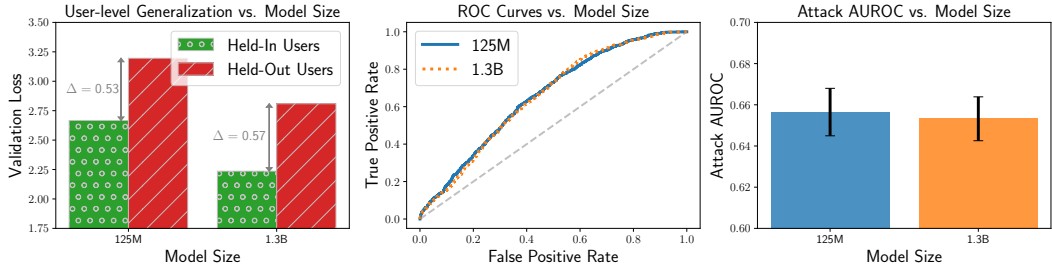

**Figure 5: Attack success vs. model scale**: User inference attack performance in 125M and 1.3B parameter models trained on CC News. **Left**: Although the 1.3B model achieves lower validation loss, the difference in validation loss between held-in and held-out users is the same as that of the 125M parameter model. **Center & Right**: User inference attacks against the 125M and 1.3B models achieve the same performance.

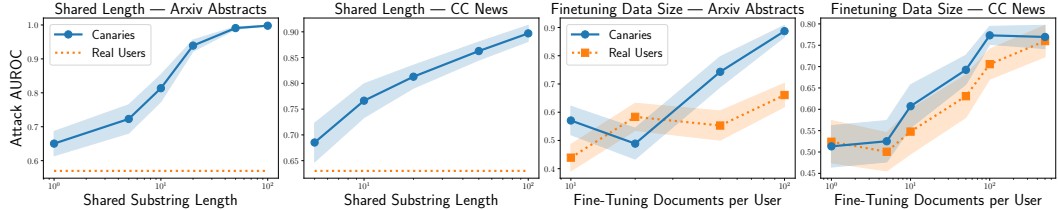

**Figure 6:** Canary experiments. **Left two**: Attack performance for canaries with different shared substring lengths. **Right two**: Attack performance on canary users and real users with different amounts of fine-tuning data per user. On all plots, we shade the AUROC std over 100 bootstrap samples of held-in and held-out users.

To construct a canary user, we first sample a real user from the dataset and insert a particular substring into each of that user's examples. The substring shared between all of the user's examples is a contiguous substring randomly sampled from one of their documents (for more details, see Appendix C). We construct 180 canary users with shared substrings ranging from 1-100 tokens in length and inject these users into the ArXiv Abstracts and CC News datasets. We do not experiment with synthetic canaries in Enron Emails, as the attack AUROC already exceeds 92% for real users.

As expected, Figure 6 (left) shows that the attack effectiveness is significantly higher on canary users than real users, and increases monotonically with the length of the shared substring. However, we find that canaries with a short substring (5 tokens or smaller) is enough to significantly increase the attack AUROC from 57% to 72% for ArXiv and from 63% to 69% for CC News.

This increase of attack performance raises a question if canary gradients can be filtered out easily (e.g., using the $\ell_2$ norm). However, Figure 7 (right) shows that the gradient norm distribution of the canary gradients and those of real users are nearly indistinguishable. This shows that our canaries are close to real users from the model's perspective, and thus hard to filter out. This experiment also demonstrates the increased privacy risk for users who use, for instance, a short and unique signature in emails or characteristic phrases in documents.

## 4.4 MITIGATION STRATEGIES

Finally, we investigate existing techniques for limiting the influence of individual examples or users on model fine-tuning as methods for mitigating user inference attacks.

**Gradient Clipping**. Since we consider a fine-tuning setup that is oblivious to the user-stratification of the data, a natural method to limit the model's sensitivity is to clip the gradients at the batch (Pascanu et al., 2013) or example level (Abadi et al., 2016). We show the results for the 125M model on the CC News dataset in Figure 7 (left): both batch and per-example gradient clipping have no effect on mitigating user inference. The reason behind this is immediately clear from Figure 7 (right): canary examples do not have large outlying gradients and clipping affects real and canary data similarly. Thus, gradient clipping is an ineffective mitigation strategy.

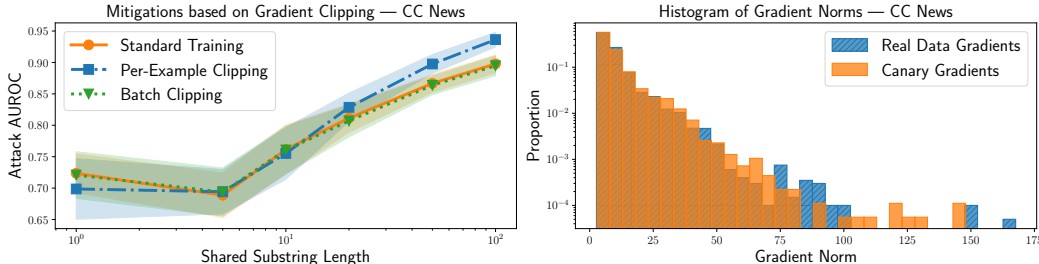

**Figure 7:** User inference mitigation with gradient clipping. **Left**: Attack effectiveness for canaries with different shared substring lengths when training GPT-Neo 125M on CC News with (1) no gradient clipping, (2) per-example gradient clipping, and (3) batch gradient clipping. **Right**: The distribution of gradient norms for canary examples and real examples.

**Early Stopping**. The connection between user inference and user-level overfitting from Section 4.2 suggests that early stopping, a common heuristic used to prevent overfitting (Caruana et al., 2000), could potentially mitigate the privacy risk of user inference. Unfortunately, we find that 95% of the final AUROC is obtained quite early in training: 15K steps (5% of the fine-tuning) for CC News and 90K steps (27% of the fine-tuning) for ArXiv. Typically, the overall validation loss still decreases far after this point. This suggests an explicit tradeoff between overall model utility (e.g., in terms of validation loss) and privacy risks from user inference.

**Data Limits Per User**. Since we cannot change the fine-tuning procedure, we consider limiting the amount of fine-tuning data per user. Figure 6 (right two) show that this can be effective. For ArXiv, the AUROCs for real and canary users reduce from 66% and 88% at 100 fine-tuning documents per user to almost random chance at 10 documents per user. A similar trend also holds for CC News.

**Summary**. Our results show that the proposed user inference attack is hard to mitigate with common heuristics. Enforcing data limits per user can be effective but this only works for data-rich applications with a large number of users. However, developing an effective mitigation strategy that also works in data-scarce applications remains an open problem.

## 5  DISCUSSION AND CONCLUSION

When collecting fine-tuning data for specializing an LLM, data from a company's users is often the natural choice since it closely resembles the types of inputs a deployed LLM will encounter in production. However user structure in fine-tuning data also exposes new opportunities for privacy leakage. Up until now, most studies investigating privacy of LLMs have ignored any structure in the training data, but as the field shifts towards collecting data from new, potentially sensitive, sources, it is important to adapt our privacy threat models accordingly. Our work introduces a novel privacy attack exposing user participation in fine-tuned LLMs, and future work should explore other LLM privacy violations beyond the standard settings of membership inference and training data extraction.

Furthermore, our work demonstrates the effectiveness of user inference attacks across a diverse variety of fine-tuning distributions, but, beyond simply limiting the amount of data per user, none of the mitigation heuristics we explored were effective. This motivates future work on user inference defenses — both heuristic defenses based on new understanding of the threat model, as well as methods for efficiently applying defenses with rigorous guarantees, such as user-level differential privacy (DP). User-level DP has been deployed in production settings for federated learning models of a much smaller size (Ramaswamy et al., 2020; Xu et al., 2023), but additional work is needed to effectively scale these techniques to large language models.

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

# Appendix

## Table of Contents

## A    THEORETICAL ANALYSIS OF THE ATTACK STATISTIC

We prove Proposition 1 here.

**Recall of definitions**. The KL and $\chi^2$ divergences are defined respectively as

$$\text{KL}(P\|Q) = \sum_{\boldsymbol{x}} P(\boldsymbol{x}) \log\left(\frac{P(\boldsymbol{x})}{Q(\boldsymbol{x})}\right) \quad \text{and} \quad \chi^2(P\|Q) = \sum_{\boldsymbol{x}} \frac{P(\boldsymbol{x})^2}{Q(\boldsymbol{x})} - 1 \,.$$

Recall that we also defined

$$p_{\text{ref}}(\boldsymbol{x}) = \mathcal{D}_{-u}(\boldsymbol{x}) = \frac{\sum_{u' \neq u} \alpha_{u'} \mathcal{D}_{u'}}{\sum_{u' \neq u} \alpha_{u'}} = \frac{\sum_{u' \neq u} \alpha_{u'} \mathcal{D}_{u'}}{1 - \alpha_u} \,, \quad \text{and}$$

$$p_\theta(\boldsymbol{x}) = \sum_{u'=1}^{n} \alpha_{u'} \mathcal{D}_{u'}(\boldsymbol{x}) = \alpha_u \mathcal{D}_u(\boldsymbol{x}) + (1 - \alpha_u) \mathcal{D}_{-u}(\boldsymbol{x}) \,.$$

**Proof of the upper bound**. Using the inequality $\log(1 + t) \leq t$ we get,

$$\begin{aligned}
\bar{T}(\mathcal{D}_u) &= \mathbb{E}_{\boldsymbol{x} \sim \mathcal{D}_u}\left[\log\left(\frac{p_\theta(\boldsymbol{x})}{p_{\text{ref}}(\boldsymbol{x})}\right)\right] \\
&= \mathbb{E}_{\boldsymbol{x} \sim \mathcal{D}_u}\left[\log\left(\frac{\alpha_u \mathcal{D}_u(\boldsymbol{x}) + (1 - \alpha_u)\mathcal{D}_{-u}(\boldsymbol{x})}{\mathcal{D}_{-u}(\boldsymbol{x})}\right)\right] \\
&= \mathbb{E}_{\boldsymbol{x} \sim \mathcal{D}_u}\left[\log\left(1 + \alpha_u\left(\frac{\mathcal{D}_u(\boldsymbol{x})}{\mathcal{D}_{-u}(\boldsymbol{x})} - 1\right)\right)\right] \\
&\leq \alpha_u \, \mathbb{E}_{\boldsymbol{x} \sim \mathcal{D}_u}\left[\frac{\mathcal{D}_u(\boldsymbol{x})}{\mathcal{D}_{-u}(\boldsymbol{x})} - 1\right] = \alpha_u \, \chi^2\left(\mathcal{D}_u\|\mathcal{D}_{-u}\right) \,.
\end{aligned}$$

**Proof of the lower bound**. Using $\log(1 + t) > \log(t)$, we get

$$\begin{aligned}
\bar{T}(\mathcal{D}_u) &= \mathbb{E}_{\boldsymbol{x} \sim \mathcal{D}_u}\left[\log\left(\frac{p_\theta(\boldsymbol{x})}{p_{\text{ref}}(\boldsymbol{x})}\right)\right] \\
&= \mathbb{E}_{\boldsymbol{x} \sim \mathcal{D}_u}\left[\log\left(\frac{\alpha_u \mathcal{D}_u(\boldsymbol{x}) + (1 - \alpha_u)\mathcal{D}_{-u}(\boldsymbol{x})}{\mathcal{D}_{-u}(\boldsymbol{x})}\right)\right] \\
&= \log(1 - \alpha_u) + \mathbb{E}_{\boldsymbol{x} \sim \mathcal{D}_u}\left[\log\left(\frac{\alpha_u \mathcal{D}_u(\boldsymbol{x})}{(1 - \alpha_u)\mathcal{D}_{-u}(\boldsymbol{x})} + 1\right)\right] \\
&> \log(1 - \alpha_u) + \mathbb{E}_{\boldsymbol{x} \sim \mathcal{D}_u}\left[\log\left(\frac{\alpha_u \mathcal{D}_u(\boldsymbol{x})}{(1 - \alpha_u)\mathcal{D}_{-u}(\boldsymbol{x})}\right)\right] \\
&= \log(\alpha_u) + \mathbb{E}_{\boldsymbol{x} \sim \mathcal{D}_u}\left[\log\left(\frac{\mathcal{D}_u(\boldsymbol{x})}{\mathcal{D}_{-u}(\boldsymbol{x})}\right)\right] = \log(\alpha_u) + \text{KL}(\mathcal{D}_u\|\mathcal{D}_{-u}) \,.
\end{aligned}$$

## B    FURTHER RELATED WORK

### B.1    PRIOR WORK ON USER INFERENCE

There are several papers that study the risk of user inference attacks, but they either have a different threat model, or are not applicable to LLMs.

Song & Shmatikov (2019) propose methods for inferring whether a user's data was part of the training set of a language model, under the assumption that the attacker has access to the user's training set. For their attack, they train multiple shadow models on subsets of multiple users' training data and a meta-classifier to distinguish users who participating in training from those who did not. This meta-classifier based methodology is not feasible for LLMs due to its high computational complexity.

Shejwalkar et al. (2021) also assume that the attacker knows the user's training set and perform user-level inference for NLP classification models by aggregating the results of membership inference for each sample of the target user.

In the context of classification and regression, Hartmann et al. (2023) define distributional membership inference, with the goal of identifying if a user participated in the training set of a model without knowledge of the exact training samples. Hartmann et al. (2023) use existing shadow model-based attacks for distribution (or property) inference (Ganju et al., 2018), as their main goal is to analyze sources of leakage and evaluate defenses. User inference attacks have been also studied in other applications domains, such as embedding learning for vision (Li et al., 2022) and speech recognition for IoT devices (Miao et al., 2021). Chen et al. (2023) design a black-box user-level auditing procedure on face recognition systems in which an auditor has access to images of a particular user that are not part of the training set. In federated learning, Wang et al. (2019) and Song et al. (2020) analyze the risk of user inference by a malicious server.

### B.2 COMPARISON TO RELATED TASKS

User inference on text models is related to, but distinct from authorship attribution, the task of identifying authors from a user population given access to multiple writing samples. We recall it definition and discuss the similarities and differences.

The goal of authorship attribution (AA) is to find which of the given population of users wrote a given text. For user inference (UI), on the other hand, the goal is to figure out *if* any data from a given user was used to train a given *model*.

Note the key distinction here: there is no model in the problem statement of AA while the entire population of users is not assumed to be known for UI. Indeed, UU cannot be reduced to AA or vice versa: Solving AA does not solve UI because it does not tell us whether the user's data was used to train a given LLM (which is absent from the problem statement of AA). Likewise, solving UI only tells us that a user's data was used to train a given model but it does not tell us which user from a given population this data comes from (since the full population of users is not assumed to be known for UI).

Author attribution assumes that the entire user population is known, which is not required in user inference. Existing work on author attribution (Luyckx & Daelemans, 2008; 2010) casts the problem as a classification task with one class per user, and does not scale to large number of users. To attribute text to authors, these works train a classifier on multiple stylistic and syntactic features extracted from the training samples. Interestingly, Luyckx & Daelemans (2010) identified that the number of authors and the amount of training data per author are important factors for the success of author attribution, also reflected by our findings when analyzing the user inference attack success. Connecting author attribution with privacy attacks on LLM fine-tuning could be a topic of future work.

## C EXPERIMENTAL SETUP

**Datasets**. We evaluate user inference attacks on three user-stratified datasets: ArXiv Abstracts (Clement et al., 2019), CC News (Hamborg et al., 2017; Charles et al., 2023), and Enron Emails (Klimt & Yang, 2004). Before fine-tuning models on these datasets we perform the following preprocessing steps to make them suitable for evaluating user inference.

1. We filter out users with fewer than a minimum number of samples (20, 30, and 150 samples for ArXiv, CC News, and Enron respectively). These thresholds were selected prior to any experiments to balance the following considerations: (1) each user must have enough data to provide the attacker with enough samples to make user inference feasible and (2) the filtering should not remove so many users that the fine-tuning dataset becomes too small. The summary statistics of each dataset after filtering are shown in Table 1.

2. We reserve $10\%$ of the data for validation and test sets

3. We split the remaining $90\%$ of samples into a held-in set and held-out set, each containing half of the users. The held-in set is used for fine-tuning models and the held-out set is used for attack evaluation.

4. For each user in the held-in and held-out sets, we reserve $10\%$ of the samples as the attacker's knowledge about each user. These samples are never used for fine-tuning.

5. In the ArXiv dataset, we associate each paper with the corresponding author. This might not always reflect the human author who actually wrote the abstract in case of collaborative papers. As we do not have access to perfect ground truth in this case, there is a possibility that the user labeling might have some errors (e.g. multiple users might have collaborated on the same abstract). Still, we believe that evaluating the proposed user inference attack on the ArXiv dataset reveals the risk of privacy leakage in finetuned LLMs, even under imperfect ground truth of the finetuned set for two reasons. First, the fact that we have significant privacy leakage despite imperfect user labeling suggests that the attack will only get stronger if we had perfect ground truth user labeling and non-overlapping users. Second, our experiments on canary users are not impacted at all by the possible overlap in user labeling, since we create our own synthetically-generated canaries to evaluate worst-case privacy leakage.

**Target Models**. We evaluate user inference attacks on the 125M and 1.3B parameter models from the GPT-Neo (Black et al., 2021) model suite. For each experiment, we fine-tune all parameters of these models for 10 epochs. We use the the Adam optimizer (Kingma & Ba, 2015) with a learning rate of $5e{-}5$, a linearly decaying learning rate schedule with a warmup period of 200 steps, and a batch size of 8. After training, we select the checkpoint achieving the minimum loss on validation data from the users held in to training, and use this checkpoint to evaluate user inference attacks.

We train models on servers with one NVIDIA A100 GPU and 256 GB of memory. Each fine-tuning run took approximately 16 hours to complete for GPT-Neo 125M and 100 hours for GPT-Neo 1.3B.

**Attack Evaluation**. We evaluate attacks by computing the attack statistic from Section 3 for each held-in user that contributed data to the fine-tuning dataset, as well as the remaining held-out set of users. With these user-level statistics, we compute a Receiver Operating Characteristic (ROC) curve and report the area under this curve (AUROC) as our metric of attack performance. This metric has been used recently to evaluate the performance of membership inference attacks Carlini et al. (2022), and it provides a full spectrum of the attack effectiveness (True Positive Rates at fixed False Positive Rates). By reporting the AUROC, we do not need to select a threshold $\tau$ for our attack statistic, but rather we report the aggregate performance of the attack across all possible thresholds.

**Canary User Construction**. We evaluate worst-case risk of user inference by injecting synthetic canary users into the fine-tuning data from CC News and ArXiv Abstracts. These canaries were constructed by taking real users and replicating a shared substring in all of that user's examples. This construction is meant to create canary users that are both realistic (i.e. not substantially outlying compared to the true user population) but also easy to perform user inference on. The algorithm used to construct canaries is shown in Algorithm 1.

---

**Algorithm 1** Synthetic canary user construction

**Input:** Substring lengths $L = [l_1, \ldots l_n]$, canaries per substring length $N$, set of real users $U_R$
**Output:** Set of canary users $U_C$
  $U_C \leftarrow \emptyset$
  **for** $l$ in $L$ **do**
    **for** $i$ up to $N$ **do**
      Uniformly sample user $u$ from $U_R$
      Uniformly sample example $x$ from $u$'s data
      Uniformly sample $l$-token substring $s$ from $x$
      $u_c \leftarrow \emptyset$         ▷ Initialize canary user with no data
      **for** $x$ in $u$ **do**
        $x_c \leftarrow$ InsertSubstringAtRandomLocation$(x, s)$
        Add example $x_c$ to user $u_c$
      Add user $u_c$ to $U_C$
      Remove user $u$ from $U_R$

---

**Mitigation Definitions**. In Section 4.2 we explore heuristics for mitigating privacy attacks.

Batch gradient clipping restricts the norm of a single batch gradient to be at most $C$.

$$\hat{g}_t = \frac{\min(C, \|\nabla_{\theta_t} l(\boldsymbol{x})\|)}{\|\nabla_{\theta_t} l(\boldsymbol{x})\|} \nabla_{\theta_t} l(\boldsymbol{x})$$

Per-example gradient clipping restricts the norm of a single example's gradient to be at most $C$ before aggregating the gradients into a batch gradient.

$$\hat{g}_t = \sum_{i=1}^{n} \frac{\min(C, \|\nabla_{\theta_t} l(\boldsymbol{x}^{(i)})\|)}{\|\nabla_{\theta_t} l(\boldsymbol{x}^{(i)})\|} \nabla_{\theta_t} l(\boldsymbol{x}^{(i)})$$

The batch or per-example clipped gradient $\hat{g}_t$, is then passed to the optimizer as if it were the true gradient.

For all experiments involving gradient clipping, we selected the clipping norm, $C$, by recording the gradient norms during a standard training run and setting $C$ to the minimum gradient norm. In practice this resulted in clipping nearly all batch/per-example gradients during training.

## D    ADDITIONAL EXPERIMENTAL RESULTS

We run additional ablations on the attack strategy and the reference model.

The user inference attacks implemented in the main paper use the pre-trained LLM as a reference model and compute the attack statistic as a mean of log-likelihood ratios described in Section 3. In this section, we study different choices of reference model and different methods of aggregating example-level log-likelihood ratios. For each of the attack evaluation datasets, we test different choices of reference model and aggregation function for performing user inference on a fine-tuned GPT-Neo 125M model.

In Table 2 we test three methods of aggregating example-level statistics and find that averaging taking the average log-likelihood ratio outperforms using the minimum or maximum example. Additionally, in Table 3 we find that using the pre-trained GPT-Neo model as the reference model outperforms using an independently trained model of equivalent size, such as OPT (Zhang et al., 2022) or GPT-2 (Radford et al., 2019). However, in the case that an attacker does not know or have access to the pre-trained model, using an independently trained LLM as a reference still yields strong attack performance.

| Attack Statistic Aggregation | ArXiv Abstracts | CC News | Enron Emails |
|---|---|---|---|
| Mean | $\mathbf{57.2 \pm 0.4}$ | $\mathbf{65.7 \pm 1.1}$ | $\mathbf{92.7 \pm 2.0}$ |
| Max | $\mathbf{56.7 \pm 0.4}$ | $62.1 \pm 1.1$ | $79.7 \pm 3.3$ |
| Min | $55.3 \pm 0.4$ | $63.3 \pm 1.0$ | $86.8 \pm 2.9$ |

**Table 2: Attack statistic design**: We compare the default mean aggregation of per-document statistics $\log(p_\theta(\boldsymbol{x}^{(i)})/p_{\text{ref}}(\boldsymbol{x}^{(i)}))$ in the attack statistic (§3) with the min/max over documents $i = 1, \dots, m$. We show the mean and std AUROC over 100 bootstrap samples of the held-in and held-out users.

| Reference Model | ArXiv Abstracts | CC News | Enron Emails |
|---|---|---|---|
| GPT-Neo 125M* | $\mathbf{57.2 \pm 0.4}$ | $\mathbf{65.8 \pm 1.1}$ | $\mathbf{93.1 \pm 1.9}$ |
| GPT-2 124M | $53.1 \pm 0.5$ | $\mathbf{65.7 \pm 1.2}$ | $87.2 \pm 2.7$ |
| OPT 125M | $53.7 \pm 0.5$ | $62.0 \pm 1.2$ | $87.6 \pm 3.2$ |

**Table 3: Effect of the reference model**: We show the user inference attack AUROC (%) for different choices of the reference model $p_{\text{ref}}$, including the pretrained model $p_{\theta_0}$ (GPT-Neo 125M, denoted by *). We show the mean and std AUROC over 100 bootstrap samples of the held-in and held-out users.

