# OpenReview forum: "User Inference Attacks on Large Language Models"
_ICLR.cc/2024/Conference — Submitted to ICLR 2024_

### Official Review · Reviewer_UDKj · 2023-10-22

**Soundness:** 3 good
**Presentation:** 3 good
**Contribution:** 2 fair
**Rating:** 5
**Confidence:** 4

**Summary:**

This paper  investigates **user inference** attacks against LLMs fine-tuned on user-stratified data. The goal is to predict if a particular user's data is used in the model's fine-tuning process. The adversary has access to a subset of a user's data and uses a simple attack that applies a threshold on the average loss of a user’s documents.  The authors also design canaries to study the influence of shared substrings. The authors also explore factors that affect the attack's success rate as well as mitigation strategies.

**Strengths:**

1.This paper first defines user-level inference attack against fine-tuned LLMs. In the threat model, the adversary’s knowledge could be different from the user’s training samples. This makes the threat model more realistic.

2.The authors demonstrate the attack's efficacy across three real-world datasets.

**Weaknesses:**

1.This paper only explores one simple threshold-based attack. Although the authors show that this simple attack works well, it is better to test other popular attacks, e.g., fitting a NN, in the membership inference literature.

2.The authors assume the documents of one user are independent (in Equation 4). This is a reasonable assumption for sample-level inference attacks, but it may not be the right assumption for user inference attacks. Consider emails, the content in email $x_{i+1}$ probably depends on the previous email $x_{i}$.

3.There are two intuitive mitigations that may be worth exploring. The first is to clip user-level gradients instead of sample-level gradients, this helps control user contribution even if one user has many more documents than others. The second is to deduplicate documents from a single user (motivated by the finding in Figure 6).

**Questions:**

The attack success rate on the Enron email dataset significantly surpasses the success rate on other two datasets. Why is this the case? Although each user contributes more documents in the email dataset, Figure 3 shows that attacks against the email dataset are still much easier even when the number of documents is limited.

---

> ### Author Response · Authors · 2023-11-17
> **Response to Reviewer UDKj**
>
> We thank the reviewer for appreciating the practicality of our threat model and for the review. We answer each of the questions below.
>
> **[Other approaches to the attack design]**
>
> State-of-the-art membership inference attacks  (e.g., LiRA by Carlini et al. 2022) require training hundreds of shadow models during the attack stage to learn the distribution of model confidences and create a test statistic. This approach would not be feasible for LLMs due to the high computational complexity incurred in training the shadow models. One of our main requirements is to design an efficient attack statistic that does not train additional shadow models. This motivates the use of the pretrained model as a reference model, resulting in the calibrated loss on the fine-tuned model normalized by the loss on the pretrained model as a test statistic.
>
> Of course, there could be other attack designs even stronger than ours with similar efficiency, but that would only make privacy leakage a bigger concern. Our attack already indicates an unacceptable level of privacy leakage in LLM finetuning and we hope that future work will address the need for developing and evaluating defenses, as well as potentially designing stronger privacy attacks.
>
> **[Independence of user’s documents]**
>
> This is a great question! In the email case, independence is a modeling assumption that, even if incorrect, makes the problem more computationally tractable. Even with this simplifying assumption, we still get an attack that is relatively strong in most cases. Relaxing this assumption could likely lead to stronger attacks, but those won't necessarily change the conclusions/observations we make in the paper.
>
> To reiterate, we do not aim to propose the strongest attacks in the paper. Rather, we focus on highlighting that even simple attacks (that are computationally efficient) can already identify privacy leakages in LLM fine-tuning.
>
> **[Mitigations: Clipping user-level gradients]**
>
> Clipping user-level gradients is a promising approach. In fact, the gold standard defense is differential privacy at the *user-level* (see Section 5), which involves clipping user-level gradients and adding noise.
>
> Unfortunately, existing LLM finetuning infrastructure does not allow for a straightforward way of implementing clipping at the user-level efficiently at scale. Our results can be seen as strong evidence in favor of (a) developing software that supports per-user clipping, and (b) augmenting training workflows to keep track of the “owner” of each data point.
>
> **[Mitigations: Deduplicate documents from a single user]**
>
> Great question! We ran some preliminary analyses on the number of duplicates in each dataset. Here are the results:
> - Nearly 10% of examples of CC News are _exact_ duplicates within users.
> - ArXiv is too large to look for exact duplicates. 2.4% of all examples in ArXiv are _approximate_ duplicates _across_ users (meaning the number the _exact_ duplicates _within users_ will likely be significantly smaller).
> - We are still in the process of analyzing the Enron email dataset and we don’t yet have final statistics.
>
> Your suggestion of deduplicating data at the user-level is a promising idea for mitigation. We will implement and evaluate it in the next revision of the paper (as this requires longer to execute).
>
> **[Attack success rate on the Enron dataset]**
>
> There are two key reasons why the user inference attack enjoyed high success on the Enron dataset, also highlighted in our analysis in Proposition 1:
> - Each user contributes a large amount of data to the training of the model. Figure 6 (right) shows that this allows for a greater attack efficacy. (Note that Figure 3 shows the effect of the attacker’s knowledge: the above result is true even when the attacker’s knowledge is minimal).
> - Second, the statistical divergence between the samples contributed by the target user compared to other training data (i.e., the KL divergence between the user’s distribution and other users’ training distributions): The larger the distance, the higher the attack success. There is more separation between users in the Enron dataset compared to ArXiv: for instance,  emails contain more user-level information in the signatures, and salutations. Other details such as addresses are also commonly found in the body of the email (see some examples [here](https://huggingface.co/datasets/aeslc))

---

> > ### Comment · Reviewer_UDKj · 2023-11-20
> >
> > Thank you for the response. I have two follow-up questions.
> >
> > Regarding clipping user-level gradients. Could the authors provide more details about the challenges of clipping user-level gradients? Given recent advances in DP training of transformers, I expect that the implementation of computing sample-level gradients is mature. If one can compute sample-level gradients, then it should also be feasible to aggregate the results to get user-level gradients.
> >
> > Regarding deduplicating documents. Thanks for the new results. What do exact/approximate duplicates mean? I'm surprised that CC News has a non-trivial duplication rate. Could you please share more details about what types of contents are duplicated? I was only expecting there are a non-trivial number of duplicates in the Enron email dataset (because of the shared signatures, as the authors have mentioned).

---

> > > ### Author Response · Authors · 2023-11-21
> > >
> > > Thank you for the follow-up questions!
> > >
> > > **User-level clipping and differential privacy (DP)**: Some of the challenges in user-level clipping are the following.
> > > - User-level clipping requires sophisticated user-sampling schemes (e.g. “sample 4 users and return 2 samples from each”). On the software side, the former requires fast per-user data loaders, which are not supported by standard workflows
> > > - This approach also requires careful accounting of user contributions per round and balancing user contributions per round and the number of user participations over all rounds. The trade-offs involved here are not well-studied and require a separate detailed investigation.
> > > - Finally, this approach can lead to a major drop in performance, especially if the number of users in the fine-tuning dataset is not very large. For instance, the Enron dataset with 150 users is too small while CC news with 3K users is still on the smaller side. It is common for studies on user-level differential privacy (= user-level clipping + noising) to use datasets with O(100K) users (e.g. Stack Overflow dataset has around 350K users).
> > >
> > >
> > > User-level differential privacy is the gold standard for protection against user inference attacks. In addition to the challenges above, this is far more compute-intensive than differential privacy (DP) at the example level, requiring aggregation of gradients from a large number of users for good performance ([McMahan et al., ICLR 2018](https://arxiv.org/pdf/1710.06963.pdf)).
> > >
> > > That being said, our results make a strong case for user-level DP. Indeed, our results motivate the separate future research question on how to effectively apply user-level DP (or simply user-level clipping) given accuracy and compute constraints.
> > >
> > > **Deduplication details**: Here are some of the details.
> > >
> > > _Exact duplicates (CC News)_: The text of the example is an exact copy of another example in its entirety. Approximately 23% of users include some duplicates, while 16% of users include more than 1 duplicate (i.e. 7% of users have exactly one duplicate). Most are regular news articles (e.g. “Mohammed Salah wins BBC African Footballer of the Year award”), and the corresponding number of duplicates is a small number like 2 or 3. Meanwhile, others are some form of web scraping errors — the following example has been repeated 88 times:
> > >
> > > ```
> > > Add the following CSS to the header block of your HTML document.
> > > Then add the mark-up below to the body block of the same document.
> > > .videoWrapper { position: relative; padding-bottom: 56.25%; padding-top: 25px; height: 0; } .videoWrapper iframe { position: absolute; top: 0; left: 0; width: 100%; height: 100%; }
> > > ```
> > >
> > > _Approximate Duplicates (Arxiv)_: The approximate deduplication approach is based on [MinHash](https://en.wikipedia.org/wiki/MinHash), a locality-sensitive hashing algorithm to find near duplicates. Each document is represented by its collection of 5-grams, which are then hashed onto integers with a hash function that has the property that more similar documents are more likely to collide. All collisions are then checked pairwise and marked as near-duplicates if their Jaccard similarity is at least 0.8.  [This blog post](https://huggingface.co/blog/dedup) outlines a similar procedure.
> > >
> > > Each example from the arxiv dataset is associated with the arxiv unique identifier (e.g. 0905.3803), so we would guess that the number of exact duplicates is extremely small.

---

> > > > ### Author Response · Authors · 2023-11-22
> > > >
> > > > Dear Reviewer UDKj,
> > > >
> > > > Thank you again for your insightful review and follow-up questions. We sincerely appreciate your time and attention to detail.
> > > >
> > > > We kindly request your attention to our most recent responses. The discussion period is closing soon, and we want to ensure that you have ample time to review our answers and provide any further feedback.
> > > >
> > > > We are eager to address any further questions or concerns you may have. Please do not hesitate to reach out if there's anything else we can do to merit a favorable evaluation. Thank you again for your time and consideration.
> > > >
> > > > Best,
> > > > The Authors

---

> > > > > ### Author Response · Authors · 2023-11-23
> > > > >
> > > > > Dear Reviewer UDKj, a gentle reminder. Please see the message above. Thank you very much for your time and consideration.

---

> > > > > > ### Comment · Reviewer_UDKj · 2023-11-23
> > > > > >
> > > > > > Thank you for the update. I appreciate the authors' attempt to incorporate the two suggested mitigations. I am maintaining my original score as the submission has not yet addressed the three points raised in my initial review. Nonetheless, I still think the idea of user-level inference attack is intriguing, and I believe this work would be stronger if it could include the results addressing any of these comments.

---

### Official Review · Reviewer_7zUW · 2023-10-31

**Soundness:** 3 good
**Presentation:** 4 excellent
**Contribution:** 3 good
**Rating:** 8
**Confidence:** 3

**Summary:**

The paper defines a threat model, called user inference, wherein an attacker infers whether or not a user’s data was used for fine-tuning. It also proposed a practical attack that determines if a user participated in training by computing a likelihood ratio test statistic normalized relative to a reference model.

The paper empirically studied the susceptibility of models to user inference attacks, the factors that affect their success, and potential mitigation strategies, using three datasets: Arxiv abstracts, CC News, and Enron Emails. Experimental results have shown that the proposed attack can achieve non-trivial performance. It also conducted canary experiments to study the privacy leakage of worst-case users. The paper also studied several mitigation methods, including gradient clipping, early stopping, and limited data per user.

**Strengths:**

- The paper introduced a new realistic threat model called user inference, and proposed a practical attack using likelihood ratio test. Experimental results show the effectiveness of the proposed attack method.
- The paper is well-written and enjoyable to read. Sufficient insights and informative figures are provided to help make the point.

**Weaknesses:**

There is no obvious weakness in the draft.

**Questions:**

1. In the current setup, the attacker is assumed to have access to a fine-tuned language model, as well as a similar reference model, which is the pretrained GPT-Neo in the experiments. If the reference model is less similar (which can be realistic, as attackers may not always have access to the base model), would it be much harder to perform the attack?

---

> ### Author Response · Authors · 2023-11-17
> **Response to Reviewer 7zUW**
>
> Thank you so much for your positive comments and suggestions! We are glad that you found the paper enjoyable to read.
>
> **[Question 1] choices of the reference model**
> That is a great question! We have performed an experiment in which we consider an attacker having access to a different pretrained reference model (Table 3 in Appendix). We obtained the best attack success when the attacker model is the same (e.g., 93.1% on Enron with GPT-Neo), but we only see a small reduction in attack success for a different pretrained model (e.g., 87.6% on OPT and 87.2% on GPT-2 for Enron). Fortunately, we found that the attack is not very sensitive to the knowledge of the reference model, and we believe that access to a public pretrained model is a reasonable assumption for the attacker.

---

### Official Review · Reviewer_TnAv · 2023-10-31

**Soundness:** 3 good
**Presentation:** 4 excellent
**Contribution:** 3 good
**Rating:** 6
**Confidence:** 2

**Summary:**

This paper proposes the threat model of _user inference_, aiming to determine whether a user’s data exists in the fine-tuning dataset using only a small set of the user’s samples. _User inference_ relaxes the assumption in previous privacy attacks, such as membership inference, which requires direct access to the specific user data from the training set. With _user inference_, the authors further investigate factors correlated with the privacy risk concerning user participation in training. They find that the uniqueness, the quantity, and the attacker’s knowledge of the user’s data can affect the attack effectiveness. This indicates how to mitigate the privacy risk using user-level differential privacy techniques during training.

**Strengths:**

The paper presents a clear problem statement and convincingly demonstrates how user inference, grounded in a more realistic assumption regarding access to user data, fills gaps in existing privacy attacks. The core idea is clearly illustrated through figures and well elaborated with theoretical and technical details in the paper.

Additionally, the paper provides thorough experiments and analysis of the attack performance across different datasets and explores factors that may influence the performance. It also offers insights into mitigating user-level privacy leakage during training. In my opinion, it sheds light on future works in privacy defenses, particularly in scaling up the user-level differential privacy for LLMs.

**Weaknesses:**

The basic assumption of user inference -- "samples from the same user are more similar on average than those from different users" -- could limit the applicability of the method. When merging samples from diverse tasks or domains during finetuning (_e.g._, blogs of different topics or on different media platforms), samples from the same domains might exhibit greater similarity than those from the same user. Also, since the pre-trained model is used as the reference model to calculate the test statistic, it’s also necessary to control and analyze the similarity among the pre-training data, the fine-tuning data, and the attacker’s knowledge, considering these potentially influential factors.

In this case, the disparity in attack performance on the three datasets may not solely be attributed to the amount of each user's data. For example, content in ArXiv Abstracts is usually more objective and its characteristics might be more reflective of the specific topics/subjects rather than the individual user identities/styles. The paper might benefit from an extended analysis or discussion on this aspect.

**Questions:**

A. Besides the amount of user data, are there other factors that could significantly influence the attack effectiveness? For example, while the unique email signatures can serve as distinct identity markers, would there be other (potentially spurious) features (_e.g._, email/article subjects) that could be captured for user identification?

B. How would the proposed attack technique perform on recent Chatbot models, such as Llama-2-chat?

---

> ### Author Response · Authors · 2023-11-17
> **Response to Reviewer TnAv**
>
> Thank you so much for your insightful comments and supportive suggestions on our work. We are glad that you found our threat model realistic and our attack a foundation for future work in privacy-preserving LLM fine-tuning.
>
> **"user inference … merging samples from diverse tasks or domains"**
>
> Great question! Note that user inference is not possible when two users have the same distribution (e.g. split a set of documents randomly into two). Intuitively, some statistical similarity between samples from the user is necessary for general user inference (not just our attack). In particular, user inference is only possible if the attacker’s i.i.d. samples are closer to the user’s fine-tuning data than to any other user’s data.
>
> This is what we meant by "samples from the same user are more similar on average than those from different users" — we have adjusted the phrasing to be more clear.
>
> **"disparity in attack performance … may not solely be attributed to the amount of each user's data [but to the text content. [Question A] other factors that could significantly influence user inference"**
>
> Great question! Indeed, the attack performance disparity on different datasets is not only due to the varying number of users (even though that is an important factor).  We identified two additional factors that contribute to the attack performance, through our analysis in Proposition 1, also confirmed experimentally,
>
> 1. _The number of documents contributed by each user_: The Enron dataset has fewer users, each contributing more samples and a larger fraction of the training set, resulting in higher privacy leakage. Figure 6 (right) shows that the attack efficacy increases with the number of fine-tuning documents.
> 1. _The statistical divergence between the samples contributed by the target user compared to other training data_ (i.e., the KL divergence between the user’s distribution and other users’ training distributions): The larger the distance, the higher the attack success. There is more separation between users in the Enron dataset compared to ArXiv: for instance, emails contain more user-level information in the signatures, and salutations. Other details such as addresses are also commonly found in the body of the email (see some examples [here](https://huggingface.co/datasets/aeslc)). On the other hand, the scientific writing style of ArXiv abstracts, which is predominantly impersonal and formal, makes the users less distinct. All these factors resulted in a more successful attack on Enron compared to ArXiv and CC News.
>
> **[Question B] proposed attack technique on recent chatbots**
>
> That is an interesting question and could be a really good topic of future research. Chatbots such as ChatGPT and even Llama-2-chat are trained with more complex pipelines, including fine-tuning on human-labeled data, training a reward model, and optimizing a policy using reinforcement learning -– quantifying the effect of each of these steps on user inference would indeed be interesting.
>
> As with any other LLM, an attacker with multiple writing samples from a target user should be able to mount an attack to extract information about the user by carefully prompting the model and using the generated text. The main issues with this approach are (a) the lack of large user-stratified chat datasets to validate the success of such an attack, and (b) the lack of software support to reproduce the multi-stage fine-tuning and alignment pipeline of chatbots. Addressing these shortcomings and running controlled user inference experiments on chatbots is an interesting direction for future work.

---

> > ### Author Response · Authors · 2023-11-22
> >
> > Dear Reviewer TnAv,
> >
> > Thank you again for your review! As the discussion period draws to a close, we kindly request that you take a moment to review our responses.
> >
> > Do you have any other questions or concerns we could address to merit a higher score? Thank you for your time and consideration!
> >
> > Best,
> > The Authors

---

### Official Review · Reviewer_hrNQ · 2023-10-31

**Soundness:** 2 fair
**Presentation:** 2 fair
**Contribution:** 1 poor
**Rating:** 3
**Confidence:** 4

**Summary:**

The goal of the paper is to evaluate the extent to which large language models leak whether any of the data generated by a user was used to fine-tune them. The authors propose a black-box attack based on the output probabilities of the model and evaluate it on three different datasets, showing their attack success to be quite high. They then study the impact of mitigations and factors such as outlier and amount of user’s data on the attack performance.

**Strengths:**

- 1) Understanding the extent to which fine-tuned LLMs leak the membership of users in a relaxed threat model (i.e., was any data of a user used to train the model) is an interesting question.
- 2) Extensive evaluation of the attack’s performance under different settings and mitigation techniques.

**Weaknesses:**

-  1) Lack of novelty: the black-box methodology used is standard and https://arxiv.org/pdf/2304.02782.pdf already proposes the same threat model. Framing existing attack terminology (“user inference”) as something new is confusing as there are already multiple works proposing the stronger threat model where either X texts of a user’s data or none were used to train the model (e.g., Song and Shmatikov). To the best of my understanding, it seems that user-level + fine-tuning + LLM is new, but the current wording doesn't reflect that, instead suggesting that the paper introduces user-level MIAs.
- 2) The arXiv abstracts dataset seems unsuitable for user-level MIAs. Indeed, the authors assign the user who submitted the paper as the “user” (i.e., the author). However, papers have multiple authors which may rewrite or edit the abstract, e.g., the PI. Hence, an abstract could be assigned as belonging to an individual even though it was largely written by someone else. I see two issues with this: (a) Potential authorship overlap between different users' documents, e.g., the same author (PI)’s data could be labeled as belonging to multiple users and (b) The task being evaluated looks like PI/research group inference attack. The “user” terminology is thus misleading, and this should be acknowledged early on.
- 3) Problematic evaluation due to non-member users’s data having already been used to pre-train the LLM. The LLMs used in this work are pre-trained in 2021 on The Pile, whose description suggests that it contains both arXiv papers and the Enron Emails dataset. The results are thus biased because the LLM already saw both members and non-members’ data in the pre-training stage. To address this, the authors could evaluate their attack on a dataset known not to have been part of training. This can be done by using arXiv abstracts from 2022 onwards (although these suffer from the problem highlighted above) or another dataset from the well-studied authorship attribution problem, see e.g., Luyckx et al. https://aclanthology.org/C08-1065.pdf)

**Questions:**

- 1) What are possible reasons for the difference in attack performance on the three datasets (Figure 2)?
- 2) Have the authors considered the connections between their work and the authorship attribution (AA) problem? One appeal of the relaxed user inference threat model is that it allows to study questions such as: what in a user’s writings (topic, style, both) is constant enough across different texts to enable the attack to succeed. These questions are widely studied in the AA literature and there are many publicly available controlled datasets, see e.g. Luyckx et al. (https://aclanthology.org/C08-1065.pdf) that this work could use in their evaluation.

---

> ### Author Response · Authors · 2023-11-17
> **Response to Reviewer hrNQ: Part 1**
>
> Thank you so much for your careful reading of the paper and your detailed comments and suggestions! We believe we have addressed all your concerns below. Please do not hesitate to reach out in case of further clarifications or comments.
>
> **Lack of novelty: black-box methodology used is standard and (Chen et al. 2023) already proposes the same threat model**
>
> Thank you for pointing out the additional reference (Chen et. al. 2023). We have improved the positioning of the paper relative to existing work on user-level privacy risks in the introduction, and related work (Section 2). While user inference attacks have been studied in the literature for other application domains such as face recognition and speech recognition, we are the first to study user inference for LLMs where the attacker does _not_ have access to the exact training samples.
>
> Specifically, **all existing work in the context of generative text models study more restrictive threat models** than what we study in this paper. Most similar to our work are the following papers:
> - [Song and Shmatikov 2019](https://arxiv.org/abs/1811.00513) study user-level inference under the stringent assumption that the  **attacker has access to the training set** of the text generation model. In contrast, our threat model assumes that the attacker may not have full knowledge of the user’s training samples, and is thus more realistic. Additionally, their attack trains multiple shadow models on subsets of the training set and a meta-classifier to differentiate users participating in training from other users. The meta-classifier-based methodology does not apply to LLMs due to the large computational cost.
> - [Shejwalkar et al. 2021](https://openreview.net/pdf?id=74lwg5oxheC) performs user inference for text-based classification models, but also assumes that the attacker has access to the user’s training samples. They do not consider generative text models in this work.
>
> Thus, to the best of our knowledge, our user inference threat model has not been considered for generative text models. This is practically relevant to many settings today, such as Smart Compose, GBoard, and GitHub Copilot.
>
> We also wish to point out that any paper published after May 28, 2023, is considered contemporaneous work as per the [ICLR policy](https://iclr.cc/Conferences/2024/ReviewerGuide). The **Chen et al. paper** you pointed out was published in August 2023 and is thus **considered contemporaneous**. We ask that you assess the merits of our work independently.
>
>
> **[ArXiv dataset] multiple authors which may rewrite or edit the abstract**
>
> We acknowledge the limitations of the ArXiv dataset you have identified. Despite them, we still believe that analyzing the attack on ArXiv shows some interesting results.
>
> While the user labeling might not be perfect, the fact that we still have significant privacy leakage suggests that the attack can only get stronger if we have perfect ground truth user labeling and non-overlapping users. Further, our experiments on canary users are not impacted at all by the possible overlap in user labeling, since we create our synthetically-generated canaries to evaluate worst-case privacy leakage. We have added a discussion in the revision in Appendix C.
>
> At a more practical level, few datasets are large enough with user annotation for evaluating our privacy attack. Thus, despite its flaws, the ArXiv dataset provides a valuable data point (in addition to our other two datasets) in the evaluation of user inference attacks.

---

> > ### Author Response · Authors · 2023-11-17
> > **Response to Reviewer hrNQ: Part 2**
> >
> > **[Q1] What are possible reasons for the difference in attack performance on the three datasets (Figure 2)?**
> >
> > Through our analysis in Proposition 1, also confirmed experimentally, we identified several factors that contribute to the attack performance.
> >
> > - _The number of users_: The larger the total number of users in the fine-tuning dataset, the more challenging the attack becomes. The attack AUROC is lowest on ArXiv which has the largest number of users (16511).
> > - _The number of documents contributed by each user_: The Enron dataset has fewer users, each contributing more samples, resulting in higher privacy leakage. This corresponds to the factor $\alpha_u$ in the theoretical analysis and is shown in Figure 6 (right).
> > - _The statistical divergence between the samples contributed by the target user compared to training data from other users_: The farther away the target user’s distribution is, the higher success the attack has. There is likely more separation between users in the Enron dataset compared to ArXiv for two reasons: (1) emails are more prone to leaking user-level information with signatures, salutations, and addresses being commonly found in the body of the email (see some examples [here](https://huggingface.co/datasets/aeslc)), and (2) The scientific writing style of ArXiv abstracts, which is predominantly impersonal and formal, makes the users less distinct. Moreover, ArXiv might have overlapping users, as you noted.
> >
> > All these factors result in greater attack success for the Enron dataset.
> >
> > **[Q2] Have the authors considered the connections between their work and the authorship attribution (AA) problem?**
> >
> > Thank you for making this interesting connection: while related, there are some key differences between the two.
> >
> > **Goal**: The goal of authorship attribution (AA) is to find which of the given population of users wrote a given text. For user inference (UI), on the other hand, the goal is to figure out _if_ any data from a given user was used to train a given _model_.
> >
> > Note the key distinction here: there is no model in the problem statement of AA while the entire population of users is not assumed to be known for UI. Indeed, UU cannot be reduced to AA or vice versa:
> > Solving AA does not solve UI because it does not tell us whether the user’s data was used to train a given LLM (which is absent from the problem statement of AA).
> > Likewise, solving UI only tells us that a user’s data was used to train a given model but it does not tell us which user from a given population this data comes from (since the full population of users is not assumed to be known for UI).
> >
> > **Approaches**: Existing approaches for authorship attribution predominantly use supervised learning, i.e., train a classifier with one class for each user. This approach does not scale to a large number of users while our attack seamlessly scales to thousands of users.
> >
> > **Datasets**: The dataset from Luyckx et al. only contains a small number of users (145) and a single example (essay) from each user, which based on our analysis makes user inference extremely difficult –- limiting the number of examples per user is our strongest defense against this attack. We are not aware of larger datasets from this area suitable for evaluating user inference. The only suitable authorship attribution dataset with more than $O(100)$ users is Project Gutenberg, which is already a part of the Pile, and shares the same evaluation concerns as the three datasets we employed.
> >
> > We added some text and references on author attribution in the extended related work section (Appendix B).
> >
> > ## Summary
> > We hope that you find the updated positioning of the paper more agreeable and that we answered all your concerns. Please do not hesitate to reach out in case of further questions or concerns. Thank you for your time!

---

> > > ### Author Response · Authors · 2023-11-22
> > >
> > > Dear Reviewer hrNQ,
> > >
> > > Thank you again for your review! As the discussion period draws to a close, we kindly request that you take a moment to review our responses.
> > >
> > > Do you have any other questions or concerns we could address to merit a higher score? Thank you for your time and consideration!
> > >
> > > Best,
> > > The Authors

---

> > > > ### Comment · Reviewer_hrNQ · 2023-11-22
> > > >
> > > > Thank you for your detailed response. I appreciate the more careful re-positioning of the paper within the related works.
> > > >
> > > > Connections to authorship attribution (AA): I wasn't referring to potential similarities between the problem statements, which I agree are different (I don't think there's a need to compare them in the paper and as far as I am concerned, the authors may remove the corresponding paragraphs from the Appendix if they wish to). What I meant is that in the AA field, researchers have explored the impact of various distributional factors on the linkability of users, curating controlled datasets which the authors may use in their evaluation. For instance: impact of number of authors (https://aclanthology.org/D13-1193.pdf), is it an author's topic or their style that makes them more re-identifiable, what happens if the target's training data is from one domain but the test data is in a different domain (https://petsymposium.org/2016/files/papers/Blogs,_Twitter_Feeds,_and_Reddit_Comments__Cross-domain_Authorship_Attribution.pdf). All of these factors are relevant to the user inference problem, since the hypothesis is that a user's data remains sufficiently consistent between the fine-tuning step and the UIA query step to enable good attack performance. Hence, my question was whether the authors looked into this literature.
> > > >
> > > > arXiv dataset: It's unclear whether results will only get stronger if the dataset was cleaned. Consider two arXiv users A and B and a paper with abstract X co-written by B but labeled as belonging to A because A submitted it. Consider an LLM fine-tuned on data labeled as A but not on data labeled as B. If the current attack method predicts that no sample from B was used to train the model, the prediction would be considered correct when it's not, inflating the performances.
> > > >
> > > > I still doubt that the datasets are clean enough for a correct evaluation of UIA risks and it seems doable to collect or curate suitable datasets with clean user labels and a variety of distribution assumptions between train and test MIA data allowing to correctly test various hypotheses.

---

> ### Author Response · Authors · 2023-11-23
>
> We really appreciate your detailed comments! In this response, we argue that:
> - the arxiv dataset are still valid, and,
> - the existing authorship attribution benchmarks are unsuitable for user inference.
>
> **Arxiv dataset**: We appreciate the reviewer’s point but we believe that the noisy labels will, on average, lead to reduced attack performance.
>
> Recall that our theoretical analysis shows that the attack is most successful when the users are well separated. Mislabeling examples leads to a mixture of data distributions (e.g. documents listed under a user A are written both by user A as well as their co-authors), which brings the data distributions closer together (this is proved below). Overall, this reduces attack efficacy as per Proposition 1.
>
> **Claim [Mixing distributions brings distributions closer]**: Let $P, Q$ be two user distributions. Suppose the mislabeling of documents leads to distributions $P’ = \lambda P + (1 - \lambda) Q$ and $Q’ = \mu Q + (1-\mu) P$ for some $\lambda, \mu \in (0, 1)$. We always have $\text{KL}(P’ \Vert Q’) \le \text{KL}(P \Vert Q)$.
>
> **Proof**: This can be seen by convexity of the KL divergence. Indeed, we have
> $$
> \text{KL}(P \Vert \mu Q + (1 -\mu)P) \le \mu \cdot \text{KL}(P \Vert Q) + ( 1- \mu) \cdot \text{KL}(P \Vert P)  = \mu \cdot \text{KL}(P \Vert Q)
> $$
> Thus, $ \text{KL}(P \Vert \mu Q + (1-\mu)P) <  \text{KL}(P \Vert Q) $.
> The same holds for the first argument by the same reasoning.
>
> We note that this argument also holds authorship attribution. If the documents are randomly mislabeled, the accuracy of any classification model only reduces the classification accuracy (see e.g. the influential paper by [Zhang et al. (ICLR 2017)](https://openreview.net/pdf?id=Sy8gdB9xx)).
>
>
> **Authorship attribution datasets**: Unfortunately, the existing curated datasets in the authorship attribution literature are **not interesting for user inference** as they are _too small_ or have _very few samples per user_.
>
> Specifically, here are the curated datasets we found and their drawbacks.
> - CCAT50 (50 authors), CMCC has (21 authors), Guardian has (13 authors) and IMDB62 (62 authors) have even fewer authors than enron, our smallest dataset, where we also have near perfect attack accuracy.
> - Project Gutenberg (29k documents and 4.5k authors) has enough users, but each user only has 6 documents on average. This is not enough to mount a user inference attack, as we need enough documents for fine-tuning (see Figure 6, right) and attacker’s knowledge (see Figure 3) for the attack results to be non-trivial.
>
> While we appreciate the reviewer’s suggestion to explore the authorship attribution datasets, we conclude that the benchmarks are unsuitable for our experiments.
>
> **Summary**: Thank you again for your time and consideration.

---

> > ### Comment · Reviewer_hrNQ · 2023-12-02
> >
> > Thank you for your reply. I acknowledge your point on the lack of public availability of large-scale AA benchmarks.
> >
> > However:
> > - there are other sources such as Reddit or Twitter that other researchers have turned to in the past to collect ground-truth user data, e.g., Song and Shmatikov use Reddit data in their user inference study. It's not clear why these were not considered in this paper.
> > - I still believe that the datasets used are not clean enough for a correct evaluation of user inference. The definition of "user" is lacking and quite loose: the paper starts with a privacy motivation (where I understand users to be people) but the evaluation strongly relaxes the user definition (with the exception of the Enron email dataset). For instance, the CCNews dataset considers the "user" of an article to be the "web domain where the article was published" which I find hard to map to "a realistic privacy attack for models trained on user data". For the arXiv dataset where ground-truth user data is available, the authors have not provided a reason why the users were not separated prior to running the attack, e.g., by ensuring that no two submitters share co-authors on their papers.
> >
> > In light of the above and after careful consideration, I have decided to maintain my score.

---

### Author Response · Authors · 2023-11-19
**Thank you for your feedback / New revision + addressing reviewer concerns**

Dear reviewers,

As the discussion period draws to a close, we kindly request that you take a moment to review our responses to your questions and comments.

In response to the comments raised by Reviewer hrNQ, we have diligently updated the positioning of our paper in relation to recent literature and significantly expanded the related work section in Section 2 and Appendix B. We believe these revisions effectively highlight our contributions relative to the existing literature – please see the blue text in the revised paper.

Furthermore, addressing the questions posed by Reviewers hrNQ, TnAv, and UDKj, we have expanded our discussion on the performance disparity observed across various datasets. Our theoretical analysis provides a comprehensive explanation for these factors, as elaborated in the detailed rebuttals provided below.

We acknowledge Reviewer UDKj's excellent suggestion regarding deduplication of the user's data. While we were unable to implement this step within the current timeframe, we are committed to incorporating it into the next revision.

If any further concerns or questions arise that we could address to improve our paper and warrant a higher score, please do not hesitate to reach out. We appreciate your time and consideration.

Sincerely,

The Authors

---

### Meta-Review · Area_Chair_jjGV · 2023-12-06

**Metareview:**

This paper delves into the risks of user data leakage in fine-tuned Large Language Models (LLMs). The authors propose a unique black-box attack based on output probabilities to assess if a user's data was used in fine-tuning. They conducted extensive evaluations across multiple datasets, revealing notable attack success rates. The strength of the paper lies in its exploration of a novel threat model in user inference and comprehensive empirical analysis. However, weaknesses include the lack of novelty in methodology, potential dataset suitability issues, and a limited exploration of mitigation strategies. The paper would benefit from clearer articulation of its novelty and a more robust dataset selection to strengthen its claims.

**Justification For Why Not Higher Score:**

The decision to reject stems from the paper's limitations in novelty and dataset suitability, as pointed out by Reviewer hrNQ. While the authors addressed some concerns, they didn't fully alleviate the doubts regarding the applicability and originality of their methodology. The unresolved issues around dataset selection and the possible overlap in user data further weakened the paper's impact, leading to its rejection despite its potential contributions.

**Justification For Why Not Lower Score:**

N/A

---

### Decision · Program_Chairs · 2024-01-16

Reject